

# Social network analysis for the assessment of pig, cattle and buffalo movement in Xayabouli, Lao PDR

Chaithep Poolkhet[1], Suwicha Kasemsuwan[1], Sithong Phiphakhavong[2], Intha Phouangsouvanh[3], Khamphouth Vongxay[4], Man Sub Shin[5], Wantanee Kalpravidh[5] and Jan Hinrichs[5]

[1] Department of Veterinary Public Health, Faculty of Veterinary Medicine, Kasetsart University, Nakhon Pathom, Thailand
[2] Department of Livestock and Fisheries, Ministry of Agriculture and Forestry, Vientiane, Lao PDR
[3] Department of Livestock and Fisheries Veterinary Vaccine Production Center, Ministry of Agriculture and Forestry, Vientiane, Lao PDR
[4] Emergency Centre for Transboundary Animal Diseases (ECTAD), FAO Regional Office for Asia and the Pacific (FAO-RAP), Vientiane, Lao PDR
[5] Emergency Centre for Transboundary Animal Diseases (ECTAD), FAO Regional Office for Asia and the Pacific (FAO-RAP), Bangkok, Thailand

Corresponding author
Chaithep Poolkhet, fvetctp@ku.ac.th

## ABSTRACT

The aim of this study is to understand the role that the movement patterns of pigs, cattle and buffalo play in the spread of foot-and-mouth disease (FMD). A cross-sectional survey consisting of a questionnaire was used in a hotspot area for FMD: Xayabouli Province, Lao People's Democratic Republic. A total of 189 respondents were interviewed. We found that the key players in this network were people who were involved with more than one species of animal or occupation (multipurpose occupational node), which represents the highest number of activities of animals moved off the holding (shown with the highest out-degree centrality) and a high likelihood of being an intermediary between others (shown with the highest betweenness centrality). Moreover, the results show that the animals moved to and away from each node had few connections. Some nodes (such as traders) always received animals from the same group of cattle owners at different times. The subgroup connection within this network has many weak components, which means a connection in this network shows that some people can be reached by others, but most people were not. In this way, the number of connections present in the network was low when we defined the proportion of observed connections with all possible connections (density). These findings indicate that the network might not be busy; only one type of node is dominant which enables increased control of disease spread. We recommend that the relevant authorities implement control measures regarding the key players, which is the best way to effectively control the spread of infectious diseases.

## INTRODUCTION

Foot-and-mouth disease (FMD) is a major infectious disease in livestock and wildlife such as cattle, buffalo, pig, sheep, goat, deer, antelope, elephant, giraffe and camelids

(*OIE, 2013*) that is caused by a virus. The etiology is a single-stranded RNA virus of the family Picornaviridae by the genus *Aphthovirus*. FMD is categorized into seven serotypes (A, O, C, SAT1, SAT2, SAT3 and Asia1) by serological and genetic heterogeneity. The infection of a susceptible host, such as cloven-hoofed animals, is mainly shown in vesicles or erosions on buccal and nasal mucous membranes, hooves, mammary glands or other infected sites of an animal's body. The disease has a high morbidity rate (possibly approaching 100% in unvaccinated animals) and a high mortality rate in young animals (more than 20% in young calves, lambs and piglets). The transmission routes of FMD are direct contact, ingestion, insemination and inhalation (*OIE, 2012*, *2013*). For laboratory diagnosis, to identify the virus in infected animals viral isolation, RT-PCR or other immunological methods are used while the serological test is performed using a method for detecting viral structural proteins or viral nonstructural proteins (*OIE, 2017*). Briefly, the risk factors of FMD occurrence and spread are low farm biosecurity (*Megersa et al., 2009*; *Ellis-Iversen et al., 2011*), the presence of high animal density in the area (*Bessell et al., 2010*), movement of infected animals (*Gibbens et al., 2001*), the presence of wildlife that are disease carriers (*Molla et al., 2010*), exposure to infected animals or secretion or products derived from infected animals (*Elnekave et al., 2016*), exposure to contaminated fomites or environments (*Alexandersen et al., 2003*) and unvaccinated animals (*Bravo De Rueda et al., 2014*).

Foot-and-mouth disease is widespread throughout the world. The disease is endemic in many parts of Asia, Africa, the Middle East and South America (*OIE, 2013*), including Lao People's Democratic Republic (Lao PDR). From 2009 to 2011, researchers identified three provinces in the northern region of Lao PDR that are a hotspot for FMD: Xayabouli, Huaphan and Xiengkhoung provinces (*Nampanya et al., 2013*). In 2011, the disease spread through 14 provinces, and the estimated financial loss at the national level was USD 102 million (*Nampanya et al., 2015*, *2016a*). At the local community level, the largest cause of financial losses was animal morbidity (*Nampanya et al., 2016b*). Therefore, the productivity and profitability of animal husbandry in Lao PDR can be improved only if FMD losses are mitigated. In this way, understanding the movement patterns of the population at risk in the hotspot zone would be valuable. For this study, social network analysis (SNA) was used to quantify the elements likely to be players in the spread of FMD.

Social network analysis is a tool that studies the relationship between the units of interest. There are two kinds of components: node and tie. Node refers to each unit of interest; tie refers to the linking of a pair of nodes (*Borgatti, Everett & Johnson, 2013*). SNA has been used to describe animal movements and trade patterns related to FMD and other infectious diseases. During an outbreak of FMD in the UK, researchers found that certain nodes (some farms, animal markets and dealers) were key players in the spread of disease during the early stage of the outbreak (*Ortiz-Pelaez et al., 2006*). All of these nodes presented a high betweenness centrality that reflected the nodes are influenced over the flow of animal movements during an outbreak. These findings were similar to the findings of another study (*Robinson & Christley, 2007*), which indicated that the auction markets in the UK were key players for the spread of disease. A study in Denmark

stated that the cattle markets influenced other nodes in the cattle movement networks (*Mweu et al., 2013*). The researchers found that the betweenness centrality (a measure of how frequent a node is located between every pair of node connections) and the closeness centrality (a measure of how close the node is to others) of these nodes were higher than for any other kind of node. In addition, in Canada, researchers described that the disease transmission was caused by the transportation of infected cattle. This network showed that the out-degree centrality (a quantification of the number of outgoing ties from the node) of the infected chain was useful for estimating the size of the epidemic in a fragmented network (*Dube et al., 2008*). In France, researchers found that the animal trade network had the characteristics of a giant strong component (GSC; *Rautureau, Dufour & Durand, 2012*). To control the disease, the researchers suggested that reducing the betweenness centrality was more efficient than breaking down the GSC. In Italy, researchers also used SNA for evaluating the potential risk to the cattle trade network (*Natale et al., 2009*). They found that the control of infectious diseases, such as FMD, should focus on animal movement activities. More recently, a study in Thailand and Cambodia found that traders and animal markets are key players in the cattle movement network, and that these important nodes should receive special attention when attempting to reduce future disease outbreaks. These researchers also found that the potential patterns of disease spread were local and long distance (*Noopataya, Thongratsakul & Poolkhet, 2015*; *Poolkhet et al., 2016*; *Khengwa et al., 2017*).

In this study, we used SNA to investigate the movement and trading patterns of pigs, cattle and buffalo in Xayabouli Province, Lao PDR. We selected this area for study because this province was identified as a hotspot for the occurrence and spread of FMD. The aim of this study was to apply SNA to better understand the behavior of the animal movement network to improve measures for disease control and prevention.

## MATERIALS AND METHODS

### Data collection and questionnaire

The cross-sectional survey in this study focused on pigs, cattle and buffalo movements in Xayabouli, Lao PRD (Fig. 1). We selected this area for research because it has been identified as a hotspot for the occurrence and spread of FMD (*Nampanya et al., 2013*). We used a questionnaire to collect from November 2013 to January 2014 after we clarified and refined the survey questions. The questionnaire comprised open and closed questions that asked respondents in face-to-face interviews and/or by phone to explain the places of origin and the destinations of animals, the people who were connected to the animal activities, the disease status of the animals, information on the animal owners and the animals' environments. All of these factors were related to the disease outbreak in the area from 2012 to 2013.

For data collection, we selected a starting location based on the advice of an officer from the Department of Livestock and Fisheries, Ministry of Agriculture and Forestry, Lao PDR. Initially, 30 interviewees were selected based on information from previous FMD outbreaks. These selected interviewees were identified as possible key players in animal movement in this area. One-wave ($k = 1$) snowball sampling was applied for additional
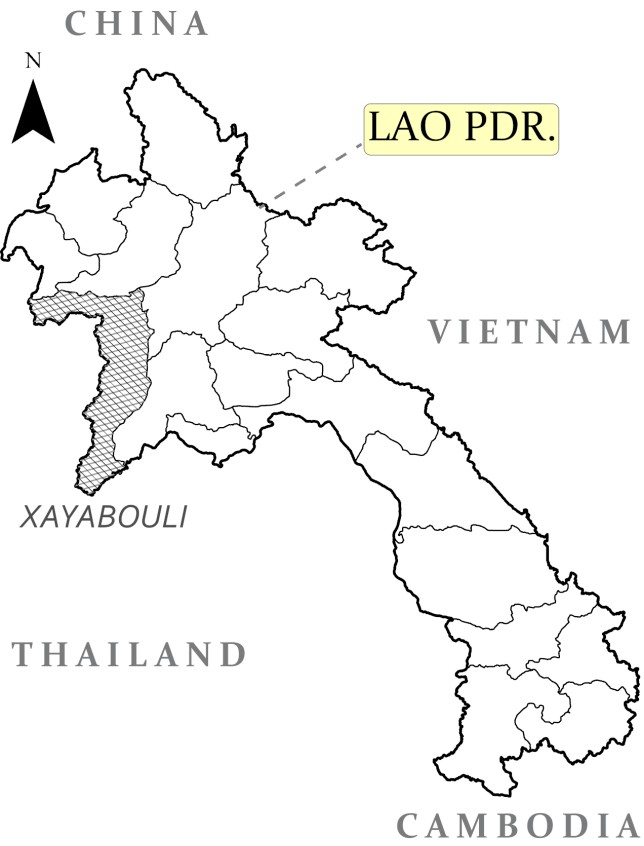

**Figure 1** **Map of the study area.** The cross-hatched area is Xayabouli, Lao PDR.

locations for data collection based on information about the starting nodes (*Wasserman & Faust, 1994*). In some locations where access was difficult, such as remote areas, a focus group was used as an additional technique for collecting data.

In this study, we paid special attention to controlling the quality of answers from respondents who knew each other or were connected in some way. We asked all interviewees whether they had ties with each other. Due to the sampling method, respondents who indicated they had a connection also had to indicate this connection. In this way, we used only matching links in the analysis. However, for the nodes outside Xayabouli, we did not validate these connections because they were outside the targeted population.

## Network properties and statistical analyses

A static, directed, weighted, one-mode network was used for the study (*Borgatti, Everett & Johnson, 2013*). The unit of interest, the node, was based on people's occupations: animal collectors, animal producers, animal traders, butchers, farm employees and slaughterhouse workers (Table 1). Some were involved with more than one animal species. For example, animal traders possibly bought and sold pigs and cattle. In this study, we specified the details of the node by animal species. In the case of people who were involved with more than one species of animal or more than one occupation,

**Table 1 Interviewees' occupations categorized into six types.**

| Occupation | Definition |
|---|---|
| Animal collector | Person who buys, fattens and sells animals for profit |
| Animal producer | Person who raises animals on any type of farm for any purpose |
| Animal trader | Person who buys and sells dead and/or live animals for profit |
| Butcher | Person who sells meat for profit |
| Farm employee | Person who works on any type of farm for wages |
| Slaughterhouse worker | Person who works in a slaughterhouse for wages |

we designated this occupation as a multipurpose activity, such as multipurpose animal producers, multipurpose butchers or multipurpose occupations. The ties between nodes were the frequency of animal movement during the study period. These movement patterns were intended for trading, restocking and reproductive purposes. In addition, if a node was connected to multiple nodes, all ties were collected. We also used local officers' expertise (a local officer is referred to staff of the Department of Livestock and Fisheries, Lao PDR) to validate the data to prevent recall bias. Uncertain ties were removed from the study after consultation with local officers.

Network analyses were performed using Ucinet 6 (*Borgatti, Everett & Freeman, 2002*). The parameters of the network were calculated at the node level. Calculated parameters were comprised of in-degree and out-degree centrality, in-closeness and out-closeness centrality and betweenness centrality. Subgroup analysis was performed using components and cut-points. The whole network structure was evaluated with the density and the clustering coefficient (Table 2; *Prell, 2012*; *Borgatti, Everett & Johnson, 2013*). In this study, multidimensional scaling was used for a visual representation of the sociograms. Mapping of Xayabouli, Loa PDR was performed by using ArcGIS 10.2.1 (ESRI, Redlands, CA, USA).

## RESULTS

Data from 189 interviewed nodes led to a total of 892 connected nodes. The mean age of the interviewed nodes was 45.7 years (median 47 years; range 20–73 years; standard deviation (SD) 11.1 years; inter-quartile range 38–53 years). Most of the interviewees were male ($n = 134$, 71%). The interviewees were involved in this network in the role of animal traders ($n = 122$, 64.5%), cattle producers ($n = 80$, 42.3%), animal collectors ($n = 76$, 40.2%), slaughterhouse workers ($n = 59$, 31.2%), pig producers ($n = 58$, 30.7%), butchers ($n = 59$, 31.2%), buffalo producers ($n = 43$, 22.8%) and farm employees ($n = 15$, 0.8%). Most of the interviewees had more than one occupation. Of the 134 nodes that raised cattle, 79 (58.9%) kept the cattle partially free, 33 (24.6%) kept the cattle in pens, 15 (11.2%) kept the cattle completely tethered and 7 (5.2%) allowed the cattle to roam freely. Of the 74 nodes that raised pigs, 69 (93.2%) were semicommercial farms, and 5 (6.8%) were small farms.

In Xayabouli, we found that the network of pig, cattle and buffalo movement was comprised of 892 nodes with 879 ties. For analyzing the node level, the mean of the in-degree centrality for all nodes was 1.4 (SD = 2.4), which is close to the out-degree centrality (mean = 1.4; SD = 2.3). The top nodes with the highest in-degree centrality

**Table 2 Description of the parameters used in this study.**

| Parameter | Description | Reference |
|---|---|---|
| Node level | | *Prell (2012)*; *Borgatti, Everett & Freeman (2002)* |
| Degree centrality | The Freeman degree centrality was used to quantify the connections of each node. In this study, two types of degree centrality were analyzed. In-degree centrality measures the number of incoming ties of the node that reflected the number of in-node movements. Out-degree centrality represents the number of animal activities that reflect off-node movement. | |
| Closeness centrality | The Freeman normalization of in- and out-closeness centrality measures the geodesic distances from a node to all remaining nodes. In-closeness centrality represents the in-node movement. Out-closeness centrality represents the out-node movement. | |
| Betweenness centrality | The Freeman betweenness represents the optimal path between all pairs of nodes. A node with a high betweenness centrality indicates good potential for the flow of animal movement in the network. | |
| Subgroup level | | |
| Component | This parameter studies how a group of nodes are connected. In this study, the component might be a weak or strong component. A strong component indicates a connection within a group of nodes in the direction of the ties whereas a weak component represents a connection in a group that disregards the direction of the ties. | |
| Cut-point | The node(s) play(s) a role in connecting the components. If (a) cut-point(s) are (is) removed from the network, then the number of components will increase. | |
| Network level | | |
| Clustering coefficient | This parameter evaluates the average number of three nodes connected together. This is a triangle of a group of nodes in the network. A network with high probability of the clustering coefficient indicates that many triangles of nodes are present there. This reflects the quantity of small clusters in the network. | |
| Density | This parameter shows the actual ties that are present in the network compared to the possible ties. Calculating the density provides the results for the probability number. | |

were the multiactivity nodes. One node represented himself as a multispecies animal collector and trader, and the four remaining top nodes with the highest in-degree centrality were also multiactivity nodes. This is similar to the top three nodes with the highest out-degree centrality. These nodes were the multiactivities nodes. The mean of the in-closeness and out-closeness centralities of all nodes were 0.1 (SD = 0.001) and 0.1 (SD = 0.001), respectively. The top nodes with the highest in-closeness centrality were the nodes that had the highest in-degree centrality. The node that had the highest out-closeness centrality was a cattle producer. The mean of the betweenness centrality was 3.5 (SD = 23.2), and the top node with the highest betweenness centrality was the multiactivities node, which had the highest out-degree centrality.

Regarding the connections among the nodes, Fig. 2 shows that most of the nodes in this network had few ties with each other. In an analysis of the subgroup, in the test of strong component characteristics, we found only a few dyads, and many components contained only one node. For the weak components testing, we found 87 components in the network (Fig. 2). The component ratios, component size heterogeneity and

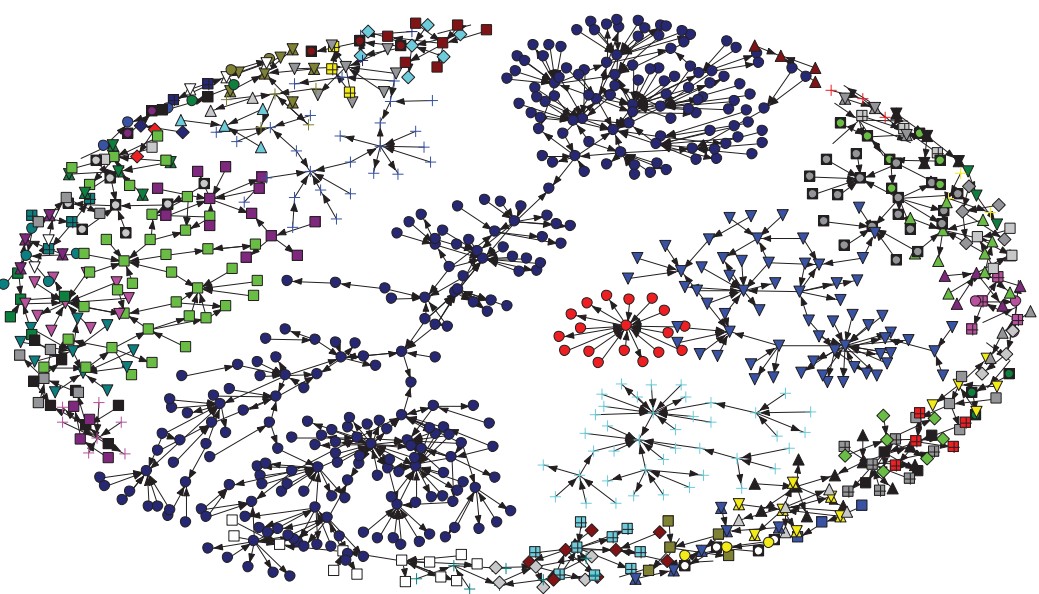

**Figure 2 The total of 87 weak components is represented by different shapes and colors for each component.** Different components represent the different sizes of the members. The largest weak component contains 297 nodes (blue circles).

fragmentation were 0.097, 0.875 and 0.876, respectively. The largest weak component contained 297 nodes, and we found that most of the remaining components contained only a few nodes. Furthermore, we found that the density for this network was 0.002. This reflects that 2% of all possible directed ties were actually present in the animal movement network. In addition, the overall clustering coefficient was 0.014. The density and the clustering coefficient indicated that this network was likely characterized by a random pattern.

We used this network to perform an analysis of the presence of FMD in Lao PDR. The results show that the disease has a scattered pattern (Fig. 3A). FMD was experienced in 30 nodes (16.04%) during the 1-year interval before the study, which consisted of multi- and single-occupation nodes with almost the same proportions. Moreover, 160 nodes answered the question about vaccination against FMD: 84 nodes (45.2%) vaccinated their animals against FMD during a 3-year period (2010–2013). In these nodes, almost all the animals were vaccinated. The sociograms according to occupation or FMD status show that cattle producers had the closest relationship to FMD compared to the other occupations (Fig. 3B). The number of accumulated ties between cattle producers and FMD was 24.

## DISCUSSION

In general, most farmers in Lao PDR raise livestock as backyard animals. The same can be said of the farmers in Xayabouli Province. This method is effective for limiting the cost of husbandry. However, this type of farming system is considered high risk for the occurrence and spread of disease—especially FMD. In this study, we found that 59.2% of farmers raised more than two animal species. For example, a farmer may raise cattle

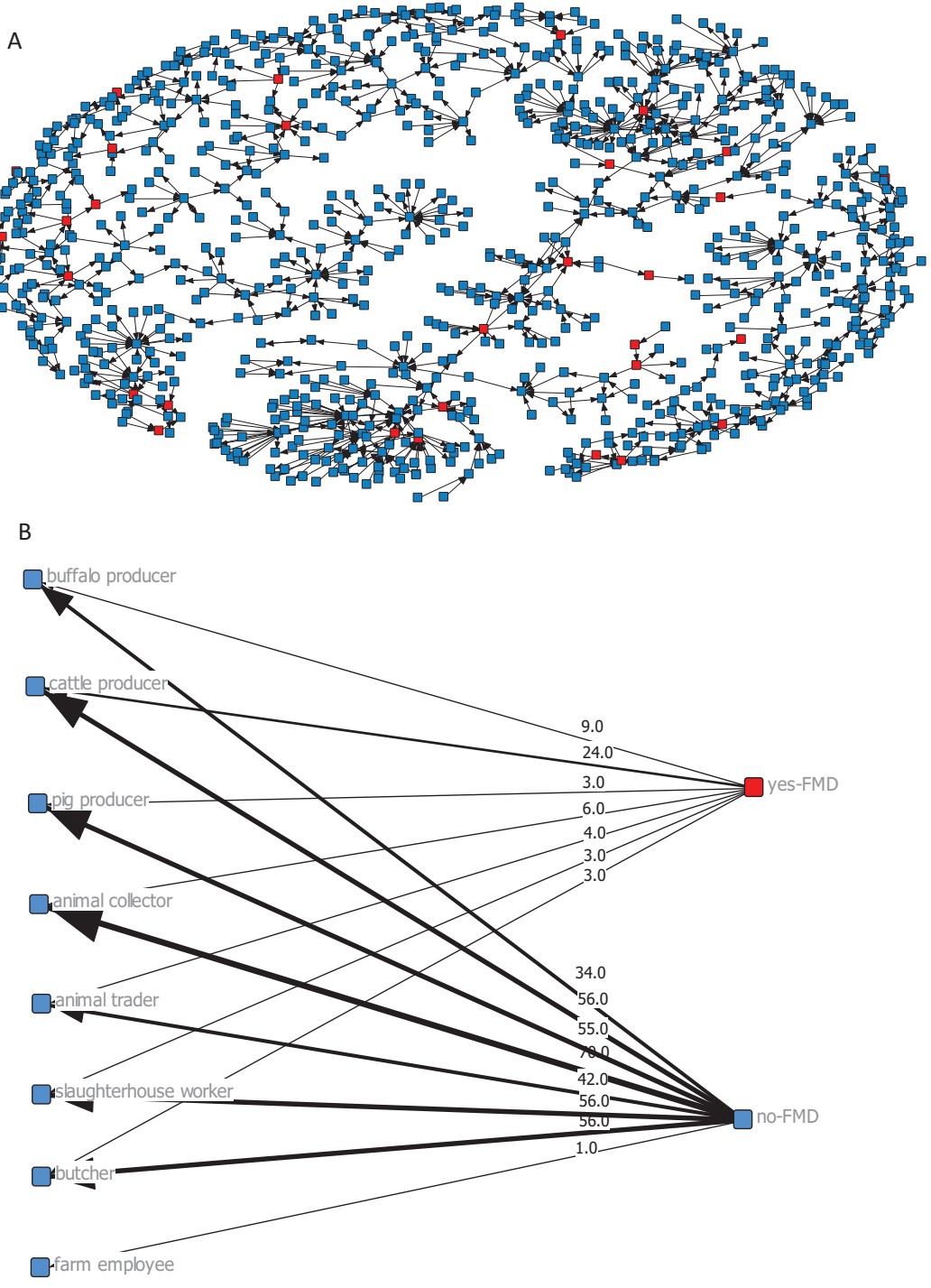

**Figure 3 (A) Sociogram of infected nodes (red squares) related to FMD and (B) sociogram of nodes according to occupation and FMD status.** The number on each tie is the number of associations of occupation and FMD status. An infected node (red square) has the highest.

and buffalo, cattle and pigs or buffalo and pigs. Most farmers raise their animals in public areas, which promotes a low biosecurity system. In our experience, most farmers are smallholders who do not pay much attention to farm biosecurity. Therefore, the Lao

authorities should routinely advise farmers about biosecurity for animal husbandry. In addition, there is a lack of access to some types of animal care, such as vaccinations, which is a big problem for the farming system in Laos. In this study, the vaccination rate for the interviewed nodes was nearly 50%. In addition, the percentage of vaccine coverage in each node was questionable. Moreover, it is possible that noninterviewed nodes have lower vaccination rates than those of the interviewed nodes, especially people in this network who are in remote areas. This is point of concern for this network regarding the occurrence and spread of FMD. An international organization should not only provide a permanent facility for the farmers in this region but also provide regular and consistent veterinary services to farmers.

The results show that the average number of in-degree and out-degree centralities was only 1.4. This means that the animals that moved to and away from each node had few connections. Therefore, when FMD occurred in an area, there were few ways for the disease to be transmitted from one node to another, making it easier for Lao authorities to control the spread of disease. The results were correlated with farmer activities, which showed that the nodes always connected to the same nodes. For example, an animal trader always received animals from the same group of cattle owners at different times; then, the trader always sent the animals to the same slaughterhouse. Lao authorities could use this information to develop a database for disease control and prevention. This database could maintain records on the place of origin and the owner (and previous owners) of each animal. This database would allow the authorities to trace the movement of each animal and would help relevant authorities understand and effectively implement control measures for the target node.

In this study, the nodes with the five highest in-degree centralities were animal collectors. All had other careers related to livestock, such as animal raising, animal slaughtering and/or animal trading. The nodes in this case were the persons who were associated with multipurpose animal handling. This means they were involved in many animal-related activities compared to other types of nodes. When disease occurred in these individuals' areas, the nodes were the most likely to transmit it to others. The relevant authorities should provide these individuals with the necessary biosecurity knowledge, attitudes and practices to decrease the spread of disease (e.g., FMD). However, once one of these nodes is infected, their behavior must be analyzed. For example, the node with a role in a slaughterhouse might have a control measure different from that of a node who raises animals. For out-degree centrality, most of the nodes were preharvest nodes. It is reasonable to assume that they had high animal move-off. A previous study indicated that infection chains are correlated with out-degree centrality (*Dube et al., 2008*). Consequently, the nodes with high out-degree centrality are spreaders of disease and are likely to increase the size of an epidemic. To prevent these nodes from becoming FMD spreaders, Lao authorities should routinely provide the nodes with a vaccination program. This could help all the members in the network minimize losses from the spread of disease. These types of nodes should be the first priority for FMD control in all endemic countries. It is well-known that the use of an emergency vaccine for reducing the spread of diseases (e.g., the ring vaccination for FMD) is necessary in

some cases (*OIE, 2012*). The present SNA results suggest that vaccinating nodes linked with patterns of animal movement is more efficient. However, this method requires data to identify nodes in the movement network, as we discussed above.

This study also showed that the connected nodes in the network had broad links. Some nodes were linked to other provinces in the country, such as Luang Prabang, and some were connected to Thailand. It would be very difficult to apply a vaccination program over wide areas during an outbreak of disease. Other control measures, such as quarantines, must be considered in these situations. Regarding the closeness centrality, the in-closeness and the out-closeness were not very high. Both parameters reflect a node's ability to easily connect animal movements in a network. This means that if disease occurs in one area, the disease will not spread too fast. The betweenness centrality was much higher overall. This means that most nodes in this network might be connectors of animal movements between a pair of connections. Equally, if the network becomes infected, then these nodes might be great connectors of disease transmission as well. Therefore, Lao authorities should be concerned about this behavior for future outbreaks. In this study, a multispecies collector who had several jobs (trader, slaughter and butcher) had the highest out-degree centrality and betweenness centrality and therefore, might be a key player for this network. These findings correspond to a previous study we conducted in Cambodia (*Poolkhet et al., 2016*).

The network comprises many weak components. This is good for disease control because it is easier to eliminate fewer links between infected and noninfected nodes. Control measures are potentially more effective for weak components than for strong components (*Dube et al., 2009*). We found a weak giant component containing 297 nodes. This was different from a previous study in Argentina (*Aznar et al., 2011*), which presented a cattle movement network that contained a GSC. A study in France indicated that a cattle trade network presented a GSC (*Rautureau, Dufour & Durand, 2012*), and in Uruguay, researchers found that cattle movement appeared in both types of components (*VanderWaal et al., 2016*). If the nodes in Lao's animal movement network increase in the future, the GSC could become an issue. Therefore, the control measures would have to be changed for further implementation.

In the present study, we also analyzed cut-point parameters but did not find any nodes with this role, because no node had the ability to connect with each component. The density and the clustering coefficient were low. These parameters indicated that the nodes in the network were connection in a random pattern. Therefore, if disease spread in this network, predicting the pattern of disease spread would be difficult. Thus, the best way to control infection in this network would be to focus on key players and control their animals' movements.

The results of this analysis reinforced our confidence in encouraging Lao authorities to implement a targeted surveillance system for FMD and if possible, the control measures we suggested. When we mapped the network with FMD status during 2012 and 2013, we did not find any interesting patterns (Fig. 3A). When we asked the local authorities, they gave us conflicting information. It is possible that when we asked sensitive questions, such as disease status, some of the respondents were unwilling to

provide accurate information. This is possibly a bias of this study. However, when we asked a different kind of question (e.g., one related to network building), we tried to control bias by including documents, such as trade documents and animal movement documents, and the experience of local authorities. We are very confident that the network data are sufficient to provide a solid interpretation. Figure 3B presents the sociogram of the node by occupation and FMD status. By focusing on infected nodes and their connections, we found that cattle producers were close to an infected node. This result indicates that cattle producers likely play an important role in the spread of disease. Moreover, some cattle producers were multipurpose occupational nodes. For example, the cattle producers were animal producers and animal collectors, traders or slaughterhouse workers during the same period. However, only a few of these individuals experienced FMD during the study period. We also found that some of the infected nodes included cattle producers who raised other types of animals as well. This behavior might be the cause of infection across animal species. Thus, this might explain the random disease patterns because different animal species have different movement patterns.

## CONCLUSIONS

The purpose of this study was to provide the government of Lao PDR a clear method for controlling FMD if the disease occurs in that country. The knowledge gained with these data will also help authorities understand the patterns of animal movements that could lead to the spread of disease. Surveillance systems should be applied to detect FMD. This study also shows how to study and understand the nature of animal movements and how to plan to control the spread of disease in developing nations.

## ACKNOWLEDGEMENTS

We would like to thank the personnel of the Department of Livestock and Fisheries, Ministry of Agriculture and Forestry, Lao PDR, for their cooperation. We would also like to thank Soulisack Panyanouvong and Somchanh Butta for their valuable help.

### Funding

This work was supported by the Government of the Republic of Korea (GCP/RAS/283/ROK) and the European Union (OSRO/RAS/901/EC). The funders had no role in study design, data collection and analysis, decision to publish, or preparation of the manuscript.

### Grant Disclosures

The following grant information was disclosed by the authors:
Government of the Republic of Korea: GCP/RAS/283/ROK.
European Union: OSRO/RAS/901/EC.

## Competing Interests

The authors declare that they have no competing interests.

## Author Contributions

- Chaithep Poolkhet conceived and designed the experiments, performed the experiments, analyzed the data, contributed reagents/materials/analysis tools, prepared figures and/or tables, authored or reviewed drafts of the paper, approved the final draft.
- Suwicha Kasemsuwan conceived and designed the experiments, performed the experiments.
- Sithong Phiphakhavong performed the experiments.
- Intha Phouangsouvanh performed the experiments.
- Khamphouth Vongxay performed the experiments.
- Man Sub Shin performed the experiments.
- Wantanee Kalpravidh performed the experiments.
- Jan Hinrichs conceived and designed the experiments, performed the experiments.

## Ethical Statement

All of interviewees were fully informed about the purpose of this study and methodology. For data gathered during the field studies, personal information was removed from the data before analysis.

## Data Availability

Raw data is provided in the Supplemental Files.

## Supplemental Information

Supplemental information for this article can be found online at http://dx.doi.org/10.7717/peerj.6177#supplemental-information.

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
