# Peer review of "Social network analysis for the assessment of pig, cattle and buffalo movement in Xayabouli, Lao PDR"

_PeerJ, doi:10.7717/peerj.6177_

## Round 0.1 · original submission · Major Revisions

Overall, both reviewers found the experimental design and analysis to be performed well, novel for the geographic region, and important for disease transmission modeling. However, both reviewers note a need for significant improvement in the language of the paper, reducing jargon, expanding definitions, and broadening the language in the abstract, introduction, and discussion to a biology audience. I second all of these suggestions and will look for a good faith effort to address them in a revision.

Please work through the reviews carefully to address the need for clarity in some places, including the methodology, and more clear language about the modeling approach for a wider audience that may be unfamiliar with network models. These reviews are well crafted and show interest in the study. I share that interest and look forward to your revision.

·

Basic reporting

Overall this manuscript needs to be vastly improved in terms of basic reporting for this general biology journal. The use of jargon related to SNA needs to be dramatically reduced, and more literature could do with being included on what previous studies have found out about FMD and what interventions can work in what types of networks to justify why finding out the network is important. Specific comments:

Whilst the abstract is well written scientifically, I am concerned about the extensive use of field specific jargon, which will reduce the ability of those not familiar with social network analysis to actually understand the paper and the meaning of the results. Without knowing the method & the term definitions the abstract is in fact currently completely incomprehensible to a general audience. Given that this is a general journal, I think the abstract should be re-written in a way to make the results understandable to people in other fields; especially as this work is applicable to veterinarians, farmers, policy makers etc.

The introduction contains extremely minimal information, obscured by extensive use of jargon and is not sufficient at all. I would like to see a bit more (something at least) about FMD, what viruses cause it etc, and a mention of how highly contagious and destructive it is, what the clinical signs are and how it is diagnosed etc. to set the scope for the reader. Further, there needs’ to be some mention of what is known about the risk factors for transmission of the disease from other countries. You mention the SNA has been used for FMD in the UK but do not say anything further; how useful was it? What did it reveal that justified your use of it now? Also, please define your terms, I know what a ‘node’ is but a generalist audience won’t, so define it the first time you use it in line 48. Again, the jargon used here (betweenness centrality and the closeness centrality of these nodes) is obscuring actual meaning. What does this mean: “the out-degree centrality of the infected chain of nodes was similar to a fragmented network”, it’s ok to say it, but then you need to explain it.

The lines in the beginning of your discussion (165-168) would be good to know in the Introduction.

Experimental design

Mostly the design of this study seems sound to me, I only have one query which I'd like to see cleared up to provide sufficient detail to replicate it: your sampling method needs explaining better. It’s not exactly clear at the moment what you did to identify 892 nodes from interviewing 189 people, especially if you required each node to reciprocally mention the other…which wouldn't be possible.

Validity of the findings

The data has been reported well and the conclusions appear to be relevant, but could do with explaining further to non-SNA specialists.

Additional comments

I think this is a valuable study, especially for the people of the province. However, further work is required to make this publishable to a standard that allows it to be understood by a general audience. Specific additional comments below:

Line 156 needs rewording, ‘They consisted of both' would make sense, but as it stands it sounds confusing to say ‘They were both'.

Line 174: you should link this statement with data from your study that shows over half of your interviewed farmers did not vaccinate against FMD

Paragraph at 178: this is a good paragraph, you have explained what your results mean to the reader and in practical terms. This is the sort of information/language required in your abstract.

This section needs expanding: what are the nodes with high betweenness centrality? Why do they influence the spread of FMD? “The betweenness centrality was much higher overall. This means that the nodes greatly influenced the spread of FMD. The nodes with a high value of betweenness centrality should be given special consideration.” Your sort of explain it in lines 224-226, but not early enough to justify your conclusion in line 221-223.

Line 233: define GSC before using the acronym.

Figure 2: different colours or shades of the same colour would help interpreting this. Also, the figure legend does not adequately explain the figure to allow a reader to interpret it if they have a general background. Please expand it to contain more detail.

Both above points also apply to Figure 3 A and B.

Reviewer 2 ·

Basic reporting

This study is a nice application of social network science for understanding infectious and emerging diseases. I would expect that studies like this should or will become more common in the future due to usefulness for understanding disease transmission. Therefore, there would be a potentially broader audience. Due to the broader audience it would be useful if the authors defined the social network metrics (in-degree centrality etc.) and gave examples. This would allow a broader audience outside of social network scientists to understand the results and use the information. The intent is to make this work relevant to a broader audience.

Experimental design

The experimental design is appropriate. More information should be given on how the people (nodes) were chosen to be interviewed and samples sizes for each category or type of nodes.

Validity of the findings

The finding are valid, however, it would be useful to better define or describe what the findings mean. For example, line 178 can better describe what it means that animals move to and away from each other. Because the metrics do not have units, it is more difficult to describe the meaning of a result and extrapolate outside of the study. Line 183 is good with giving the exact example of what the results means.

Again, describing what in-degree centrality means in social network science will be helpful to a broader audience.

Additional comments

Line 35 Define more narrowly multi-species. For example, ungulates and not zoonotic.
Line 69 Add to Aims something to briefly describe the data collection and focus group.
Line 76 Questionnaires sent to who and how were they selected?
Line 131 It might be useful to have a table that describes each metric (e.g. in-dgree centrality) to a general audience.
Line 227 Define what a weak component means.
Line 238 Define cut-point parameters
Line 277 In the Ethical Statement was this research approved by an Institutional Review Board (IRB)? Lao PDR to the institutions may not have an IRB but if they do then this should be included.
Table 1 Can easily add the sample size for group for interviews
Figure 1 Could be decreased in size or zoomed in to just include Lao PDR
Figure 2 and Figure 3A are so similar it difficult to see the difference. One could be omitted to the clarity increased.
Figure 3 requires cleared and larger font and the numbers and units described in the legend.

---

## Round 0.2 · accepted · Accept

Thank you for your effort to review and strengthen the paper using the comments of two qualified reviewers. The work is much improved and ready for publication, pending a revision to the map in Figure 1. I am not comfortable with using only country labels without borders. While it is important to indicate Laos, it is also necessary to place it in context. Please review Figure 1 to include grayed out surrounding neighbors. If you use a light gray fill and two shades darker gray country boundaries, the map will have more meaning. This can be fixed in production.

Your efforts to address the reviewers comments seriously is appreciated and the paper stronger for it. Thank you. Congratulations.

#